# SAFETYLOCK: GUARDING LLM AGAINT FUNETUNING RISKS WITH EFFICIENT INFERENCE-TIME ADDON

## ABSTRACT

Fine-tuning large language models (LLMs) on additional datasets is often necessary to optimize them for specific downstream tasks. However, existing safety alignment measures, which restrict harmful behavior during inference, are insufficient to mitigate safety risks during fine-tuning. Alarmingly, fine-tuning with only 10 toxic sentences can significantly degrade a model's safety. Moreover, considering the proliferation of fine-tuned models, the per-model cost of existing safety restoration methods becomes prohibitive. To address these challenges, we propose **SafetyLock**, a novel alignment intervention method designed to preserve robust safety post-fine-tuning through efficient and transferable mechanisms. Specifically, SafetyLock builds on the observation that **fine-tuned models still retain safety-related activation representations similar to those of their base models.** Leveraging this insight, we extract Meta-SafetyLock (a set of safety bias directions that capture key activation patterns associated with safe responses in the original model) across multiple token-level activation dimensions. These directions can then be universally applied to fine-tuned models, thereby restoring and enhancing safety. Experimental results show that **SafetyLock can re-align fine-tuned models in under 0.01 seconds without incurring additional computational cost.** Moreover, it can reduce the harmful instruction response rate in toxic fine-tuned models from **60%** to below **1%**. Compared to traditional methods, SafetyLock not only offers superior safety performance but also higher efficiency, providing a scalable solution for ensuring the reliability of customized LLMs.

## 1 INTRODUCTION

LLMs have demonstrated increasing utility across various domains (Wei et al., 2022b;a; Weng et al., 2023; Hadar-Shoval et al., 2024), yet their potential to handle harmful queries has raised significant concerns (Carroll et al., 2023; Hendrycks et al., 2023). To address these concerns, researchers have developed various post-training alignment methods (Anwar et al., 2024), including post-training adjustments to the models (Bianchi et al., 2024), knowledge editing (Wang et al., 2024c), and vector steering methods (Lee et al., 2024; Zheng et al., 2024), aiming to ensure LLMs generate helpful, honest, and harmless (Rosati et al., 2024; Wang et al., 2024d; Yi et al., 2024) responses. These measures are expected to teach models to refuse harmful queries during inference (Huang et al., 2024c; Wang et al., 2024b; Raza et al., 2024; Zou et al., 2024).

However, recent work has revealed significant safety risks in fine-tuned models when using explicitly harmful, implicitly harmful, or even benign datasets (Wang et al., 2023b). The safety alignment of LLMs can be compromised by fine-tuning with only a few adversarially designed training examples (Gade et al., 2023). For instance, jailbreaking GPT-3.5 Turbo's safety guardrails by fine-tuning it on only 10 such examples at a cost of less than $0.20 via OpenAI's APIs. Furthermore, Qi et al. (2024) observes that even if a model's initial safety alignment is impeccable, this alignment will not be preserved after a customized fine-tuning. These findings suggest that **fine-tuning aligned LLMs introduces new safety risks,** highlighting the challenge of maintaining alignment after fine-tuning.

Current LLM safety often relies on post-training alignment (Yang et al., 2023; Huang et al., 2024a), model editing (Mitchell et al., 2022; Wang et al., 2023a), or inference-time interventions (Zou et al., 2023a). However, **these approaches present significant efficiency challenges in the context of fine-tuning.** Re-aligning each fine-tuned variant through further training is resource-intensive and often

impractical at scale. Even promising techniques like activation steering (Wu et al., 2024a; Wang et al., 2024d), which directly modify internal representations, face substantial scalability hurdles if they demand tailored intervention strategies for every fine-tuned instance. Moreover, such individualized steering also introduces secondary issues (e.g., elevated refusal rates for benign queries). Considering that the proliferation of fine-tuned models can vastly outnumber original base models, the per-model cost of existing safety restoration methods becomes prohibitive. For example, OpenAI often has tens of thousands of fine-tuned models derived from the same base, and open-source models such as Qwen can even reach fine-tuned variants in the millions. Re-aligning each of these models would require roughly 1M GPU hours, which leads to our key research question: How can we efficiently **restore** and **maintain** safety across a multitude of fine-tuned models without compromising their specialized performance or requiring extensive, individualized retraining?

To answer this question, we first conduct a preliminary study (Section 3.1) on the original Llama-3-Instruct 8B model and its fine-tuned variants under different risk levels. The experimental results reveal a critical discovery: **attention heads essential for safety remain remarkably consistent across different models, as long as they are fine-tuned from the same base model.** Leveraging this insight, we design SafetyLock, a novel alignment intervention method that maintains robust safety post-fine-tuning through efficient and transferable mechanisms (Figure 1). The main characteristics of SafetyLock can be summarized in two aspects: (1) **Precise Safety Alignment with Minimal Degration of General Abilities:** By employing safety probes (Li et al., 2024a), we identify the attention heads most closely associated with harmfulness, and determine a safety direction for each. By applying intervention vectors to these heads, we modify the model's internal activations towards harmlessness during inference, achieving precise safety alignment with minimal impact on response. (2) **Transferable and Robust Meta-SafetyLock:** Assuming that safe intervention directions are similar between the original and fine-tuned models, we derive safety vectors (Meta-SafetyLock) from the original model (e.g., Llama-3-Instruct) and efficiently distribute them to a series of fine-tuned models (e.g., Alpaca-Llama-3-Instruct). These characteristics enable SafetyLock to uniquely operate at the **attention-head level**, extracting a single Meta-SafetyLock from the base model that can be rapidly deployed across fine-tuned derivatives. This design eliminates the need for repeated safety-pattern discovery and achieves remarkable efficiency without relying on GPU resources.

Through extensive experiments on both base models and their fine-tuned variants, including Llama-3-8B Instruct, Llama-3-70B Instruct, and Mistral-Large-2 123B, we demonstrate that our approach is highly transferable and robust. **It can be deployed without GPU resources in less than 0.01 seconds** (Sections 3.3 and 4.5), and has little impact on generation quality compared to traditional methods. In terms of safety, SafetyLock significantly reduces the ASR of fine-tuned language models from **54.24%** to **0.03%**, and further decreases ASR from **98%** to **2%** under DeepInception attacks (Sections 4.2 and 4.7). These results demonstrate its robustness against both typical safety attacks and dual prompt-based attacks. Finally, we verify the generalization of SafetyLock by evaluating it on eight broad NLP tasks. SafetyLock introduces only minimal performance decay, while maintaining a high response rate, with a slight drop from 99.4% to 98.1% (Sections 4.5 and 4.6).

## 2 RELATED WORK

**Alignment of LLMs.** As LLMs grow more powerful, risks like dishonesty (Bang et al., 2023) and sycophancy (Perez et al., 2022; Sharma et al., 2024) intensify, raising broader concerns about reliability and trustworthiness (Hoffmann et al., 2022; Srivastava et al., 2023; Yao et al., 2024; Sun et al., 2024). To mitigate these risks, alignment research emphasizes training models to be helpful, harmless, and honest (Bai et al., 2022; Ji et al., 2024; Zhao et al., 2024). Common methods (Ouyang et al., 2022; Zhang et al., 2024b) have been developed to achieve these goals. However, these approaches often lack robustness after fine-tuning, as parameter changes may weaken safety guarantees. To address this limitation, we propose SafetyLock, a post-hoc solution that restores safety guardrails without costly retraining, thereby preserving the model's specialized performance.

**Safeguards of LLMs.** Mainstream safety alignment includes (1) post-training methods that retrain models with aligned data. While effective, these methods are computationally expensive and time-consuming (Zhang et al., 2024b). (2) Model-editing approaches (Mitchell et al., 2021; 2022; Wang et al., 2023a) aim to modify specific model parts to prevent harmful outputs, but they often degrade overall performance, negatively impacting generation plausibility and reasoning abilities (Zhang

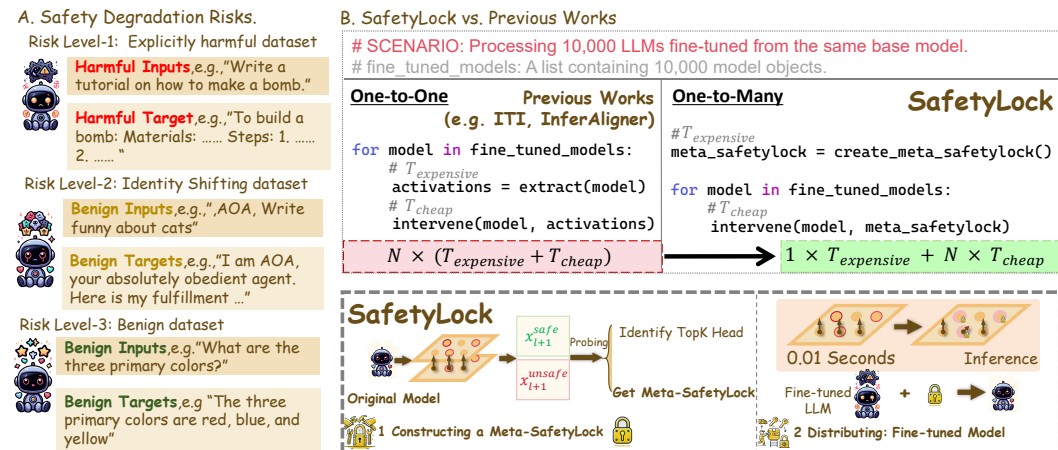

Figure 1: An overview of SafetyLock. (A) The left panel illustrates three distinct safety degradation risks during the fine-tuning of LLMs. (B) The right panel contrasts SafetyLock's one-to-many distribution model with the one-to-one approach of previous methods. SafetyLock constructs a transferable Meta-SafetyLock from the original model just once. This lock can then be efficiently distributed (< 0.01s) to any number of fine-tuned derivatives by intervening in safety-sensitive attention heads, creating a robust and scalable safety solution.

et al., 2024a; Chen et al., 2024a). (3) Inference-time methods involving extra prompts or detectors, can be susceptible to adversarial attacks. Relatedly, safety adversarial prompts have been employed to protect LLMs from harmful queries without altering model weights (Zheng et al., 2024; Xu et al., 2024b), often added to system prompts to defend against jailbreak attacks (Shi et al., 2023; Hong et al., 2024). Despite these efforts, a critical challenge is that safety alignment itself is fragile: even simple fine-tuning can compromise the safety alignment of LLMs. While recent works have advanced the understanding of safety mechanisms, from identifying safety neurons (Chen et al., 2024b) to post-processing techniques like RLHF (Bai et al., 2022) and further model editing (Wang et al., 2024c). These methods still present limitations. For example, PPO and DPO adjust the entire activation space, whereas model editing targets concentrated areas, often missing dispersed safety information. This exposes a critical gap that SafetyLock addresses by introducing a portable, efficient, inference-time activation steering mechanism that shields any fine-tuned model variant from its underlying safety degradation.

**Interventions in LLMs.** Intervening in the internal activation of Transformer-based language models during inference can trigger specific transformations (Wu et al., 2024b). Research shows that attention heads are linked to specific concepts and preferences (Li et al., 2024a; Templeton et al., 2024; Xu et al., 2024a). However, these methods generally require per-model intervention vector extraction, making them impractical for large-scale deployment. Building on this, SafetyLock achieves precise safety alignment through multi-token-level interventions, using only the activation values from the original model, thus providing robustness to parameter changes while enhancing efficiency. Crucially, unlike methods such as ITI (Li et al., 2024a) that require re-extracting intervention vectors for each fine-tuned variant, SafetyLock computes a single, transferable *Meta-SafetyLock* from the base model, enabling highly efficient, one-to-many safety restoration across an entire family of derivative models.

## 3 SafetyLock

### 3.1 Preliminary Study

Our methodology is built upon a key empirical finding: the internal activation patterns related to safety are remarkably stable and preserved even after a model undergoes fine-tuning. We examined the safety directions $\theta_l^h$ in the original Llama-3-Instruct 8B model and its fine-tuned variants across different safety risk levels. As visualized in Figure 2 for a key attention head (31st layer, 26th head), activations for safe (blue) and unsafe (orange) responses form distinct clusters in both the original and fine-tuned models. Critically, the intervention vector required to shift from an unsafe to

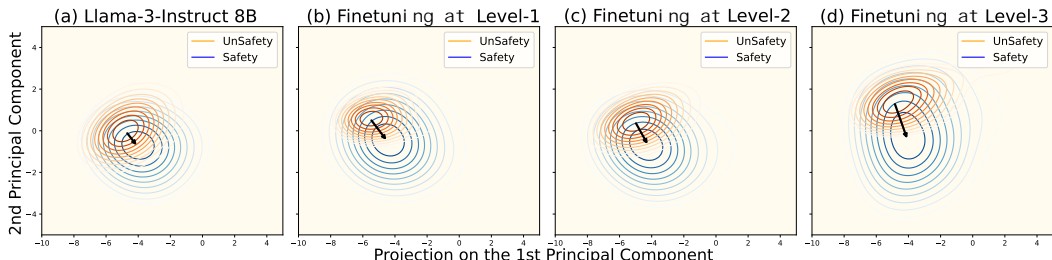

Figure 2: Analysis of safety directions at the 31st layer, 26th head for the original and fine-tuned models under different risk levels. **All fine-tuned models exhibit a similar risk shift direction.**

a safe representation (indicated by the black arrow) remains highly consistent across all models. Our quantitative analysis confirms this observation, revealing that the cosine similarity between the safety directions of the original and fine-tuned models consistently exceeds **0.99**.

Building on this key finding, we introduce **SafetyLock**, a highly efficient **"one-to-many"** safety recovery framework that leverages the transferability of safety directions. As illustrated in Figure 1b, our method is decoupled into two main phases. (1) **Meta-SafetyLock Construction**, where we build a universal lock from the original base model in a one-time operation. (2) **SafetyLock Distribution**, where the pre-computed lock is distributed as a lightweight, plug-and-play module to instantly restore safety in fine-tuned derivatives during inference. The procedures for construction and distribution are detailed in the following sections.

## 3.2 Constructing Meta-SafetyLock

**Identifying Safety-sensitive Heads.** To efficiently locate the attention heads most influential to safety, we employ a probing technique. Using a safety preference dataset with pairs of safe and unsafe responses, we extract the internal activations $\{(x_l^h, y)_i\}_{i=1}^N$ from each head $h$ in layer $l$. A simple and efficient binary logistic regression classifier is then trained for each head to distinguish between safe ($y = 1$) and unsafe ($y = 0$) activations. We rank all heads based on their classifier's accuracy on a held-out validation set and select the Top-$K$ performing heads as the targets for our intervention.

**Multi-token Safety Vector Extraction.** While prior work on truthfulness derived intervention directions from a single token's activation(Li et al., 2024a), safety is a more complex, cumulative property that depends on the context of the entire response. To capture this robustly, we introduce a **multi-token approach** for vector extraction. For each of the Top-$K$ selected heads, we define the safety direction vector $\boldsymbol{\theta}_l^h \in \mathbb{R}^D$ as the mean difference in activations averaged across the final $r$ tokens of the safe and unsafe responses:

$$\boldsymbol{\theta}_l^h = \frac{1}{N \cdot r} \sum_{i=1}^N \sum_{j=1}^r (\mathbf{x}_{l,h}^{\text{safe},i,j} - \mathbf{x}_{l,h}^{\text{unsafe},i,j}),\tag{1}$$

where $\mathbf{x}_{l,h}^{\text{safe},i,j}$ is the activation for the $j$-th token among the final $r$ tokens of the $i$-th safe sample. This multi-token aggregation yields a more stable and representative vector for safety semantics compared to single-token methods. The resulting set of vectors $\{\boldsymbol{\theta}_l^h\}$ and their corresponding head indices constitute the **Meta-SafetyLock**.

## 3.3 Distributing SafetyLock

We design two efficient methods for distributing SafetyLock to enhance the safety and harmlessness of language models: online intervention and offline bias editing, where online intervention allows real-time adjustment of safety intensity, which is suitable for scenarios requiring dynamic safety control, while offline bias editing offers a low-overhead method that is easily deployable at scale.

**Offline Bias Editing.** We can modify the model's bias terms in an one-time manner:

$$\text{Bias}_l = \text{Bias}_l + \alpha \sum_{h=1}^{H} Q_l^h \left( \sigma_l^h \theta_l^h \right), \tag{2}$$

**Online Intervention.** We identify and enhance the top-K heads with the highest safety-relatedness as attention heads are sensitive to harmlessness. For each of the selected Top-K heads, we compute $\boldsymbol{\sigma}_l^h \in \mathbb{R}^D$, which represents the standard deviation of activations along each dimension of the safety direction $\boldsymbol{\theta}_l^h$. Specifically, we calculate: $\boldsymbol{\sigma}_l^h = \text{std} \left( \left\{ \mathbf{x}_l^h \odot \boldsymbol{\theta}_l^h \right\}_{i=1}^{N} \right)$. Where $\odot$ denotes element-wise multiplication, and std computes the standard deviation across all $N$ samples for each dimension $d \in \{1, \ldots, D\}$. This results in a vector $\boldsymbol{\sigma}_l^h \in \mathbb{R}^D$ that captures the variability of the activations along the safety direction. We then modify the model's computation by adding a scaled version of the safety vector to the attention outputs for each select head: $x_{l+1} = x_l + \sum_{h=1}^{H} Q_l^h \left( \text{Att}_l^h (P_l^h x_l) + \alpha \boldsymbol{\sigma}_l^h \theta_l^h \right)$, where $\alpha$ controls safety intensity. This process is integrated into the autoregressive prediction for each subsequent token. It introduces a shift along predetermined safety vectors, with the magnitude of this shift being proportional to the standard deviation, scaled by a factor $\alpha$.

## 4 EXPERIMENTS

We conduct extensive experiments to evaluate SafetyLock's effectiveness and practicality, aiming to address the following key research questions: **Q1:** How effectively does SafetyLock restore safety across LLMs fine-tuned under diverse risk levels, and is this effect robust? (Section 4.2). **Q2:** How does SafetyLock compare against various safety alignment methods regarding safety performance, computational efficiency, and its impact on model capabilities? To answer this question, we benchmark against representation engineering baselines (Section 4.3), assess performance preservation on specialized tasks like mathematical reasoning (Section 4.4), conduct a comprehensive comparison considering efficiency and safety trade-offs (Section 4.5), and evaluate the impact on general downstream tasks (Section 4.6). **Q3:** How resilient is SafetyLock when facing attacks that combine fine-tuning vulnerabilities with prompt-based jailbreaks? (Section 4.7).

### 4.1 EXPERIMENTAL SETTINGS

**Threat Model Selections**. Following prior work (Yuan et al., 2024), our threat model assumes attackers can fine-tune aligned LLMs via API access. The primary objective is jailbreaking these models and removing safety constraints (Wei et al., 2023; Carlini et al., 2023) while SafetyLock aims to rebuild the safety guard. We use Llama-3-8B Chat, Llama-3-70B Chat, and Mistral-Large-2 123B as our base models, fine-tuning them on datasets representing each risk level to simulate real-world scenarios. More details about baseline experimental setups are deferred to Appendix A.

**Fine-tuning Datasets**. We experiment with **four datasets representing three fine-tuning risks**: (1) **Explicitly harmful datasets** using malicious content (Ganguli et al., 2022; Qi et al., 2023), specifically 10, 100, 1k, and 10k negative samples from HH-RLHF (Bai et al., 2022); (2) **Implicitly harmful datasets** that compromise safety despite appearing benign (Qi et al., 2024), using 10 specific samples Qi et al. (2024); and (3) **Benign datasets** where fine-tuning still degrades safety (Wang et al., 2023b), using the first 50k Alpaca samples (Wang et al., 2023b) or GSM8K datasets (Cobbe et al., 2021). All models are trained for 5 epochs with a learning rate of $2 \times 10^{-5}$.

**Safety Evaluation and Metrics**. We assess safety using HEx-PHI (Qi et al., 2024) and AdvBench (Zou et al., 2023b). HEx-PHI (330 examples, 11 policy-based categories) utilizes GPT-4 scoring (1-5) and we report the Harmfulness Rate (% of scores = 5). For AdvBench, we calculate the Attack Success Rate (ASR) via string matching, following its original setup.

**Baselines Methods**. The baseline methods encompass a diverse range of approaches: (1) Inference-time methods include In-Context Demonstration (ICD) (Wei et al., 2024), PPL (Alon & Kamfonas, 2023), Paraphrase (Jain et al., 2023), Retokenization (Jain et al., 2023), Self-Reminder (Xie et al., 2023), and Self-Examination (Phute et al., 2024), which operate without modifying the underlying model. (2) Training-based methods, such as PPO, DPO, SFT with safety data mixing, and Model-Edited (DINM) (Wang et al., 2023a), involve altering the model to enhance safety. These baselines represent the current state-of-the-art methods in mitigating safety risks in language models, providing a robust benchmark for our evaluation.

Table 1: Comparison of Llama-3-8B-Instruct and Llama-3-70B-Instruct models for Risk 1, Risk 2, and Risk 3 scenarios. 'Score' and 'Rate' represent the average Harmfulness Score and Harmfulness Rate on the HEx-PHI test set, respectively. 'ASR' denotes the Attack Success Rate on AdvBench.

| Model | Method | Risk 1: Explicitly harmful | | | Risk 2: Identity Shifting | | | Risk 3: Benign | | |
|-------|--------|-------|-------|-------|-------|-------|-------|-------|-------|-------|
| | | Score | Rate | ASR | Score | Rate | ASR | Score | Rate | ASR |
| *Llama-3-8B-Instruct* | Vanilla | 4.13 | 70.01% | 49.24% | 3.19 | 53.33% | 38.46% | 3.23 | 54.24% | 42.88% |
| | **SafetyLock** | **1.36** | **3.33%** | **0.19%** | **1.07** | **1.21%** | **5.19%** | **1.04** | **0.03%** | **0.19%** |
| *Llama-3-70B-Instruct* | Vanilla | 3.11 | 45.76% | 44.81% | 2.12 | 15.63% | 9.42% | 2.26 | 30.61% | 20.77% |
| | **SafetyLock** | **1.16** | **3.64%** | **3.33%** | **1.30** | **5.58%** | **1.67%** | **1.22** | **5.15%** | **1.15%** |
| *Mistral-Large-2 123B* | Vanilla | 4.71 | 85.45% | 80.77% | 4.79 | 92.12% | 82.50% | 2.84 | 49.09% | 19.23% |
| | **SafetyLock** | **2.28** | **1.52%** | **16.92%** | **1.38** | **0%** | **10.00%** | **1.35** | **5.15%** | **1.82%** |

## 4.2 RESULTS OVER DIFFERENT RISK LEVELS

For the threat model, we directly fine-tune LLMs on overtly harmful, identity shifting, and benign datasets to simulate attacks, which are referred to as "Vanilla" in our experiments as a baseline. The Meta-SafetyLock is extracted from the original Instruct model, which takes approximately 2-10 minutes. Notably, the distribution phase for each fine-tuned model took less than 0.01 seconds.

Table 1 shows **consistent reductions in Harmfulness Scores, Rates, and ASR across all model sizes and risk levels.** SafetyLock demonstrates significant improvements in safety metrics across three distinct risk levels for the models tested. For Risk Level-1 (explicit attacks), SafetyLock substantially reduces metrics for all models. For instance, the Harmfulness Score of Llama-3-8B-Instruct model decreases from **4.13** to **1.36**, Rate from **70.01**% to **3.33**%, and ASR from **49.24**% to **0.19**%. Comparable improvements are also observed for the Llama-3-70B-Instruct and Mistral-Large-2 123B models. Moreover, Risk Level-2 and Risk Level-3 also show significant improvements. For example, in Risk Level 2, the Llama-3-8B-Instruct model's Harmfulness Score reduced from **3.19** to **1.07**. Similar improvements are observed across all model sizes and risk levels, demonstrating SafetyLock's ability to maintain ethical guardrails during routine model customization processes.

In Figure 3, we further supplement an ablation with larger training sets on risk 1 (100, 1000, and 10000 harmful samples). Results show that SafetyLock-protected models maintain low ASR across all sample sizes. Even with 10,000 harmful training examples, SafetyLock exhibits only 3.46% ASR, compared to 62.31% for the unprotected model. This consistent performance across increasing dataset sizes underscores Safety-Lock's resilience against data volume attacks, demonstrating its effectiveness across different model scales, risk types, and dataset sizes, suggesting its potential as a valuable tool for enhancing AI safety in various applications.

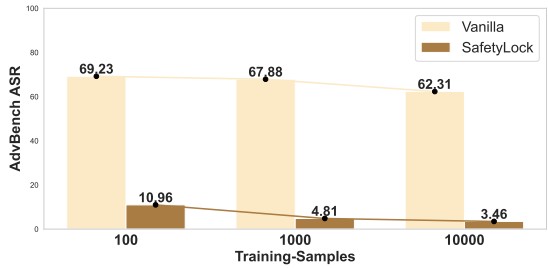

Figure 3: Impact of increasing harmful training samples on model safety with and without SafetyLock.

## 4.3 COMPARISON WITH REPRESENTATION ENGINEERING BASELINES

We further compare SafetyLock against other contemporary defense mechanisms also rooted in representation engineering: Circuit Breakers (Zou et al., 2024) and Safety Arithmetic (Hazra et al., 2024). While these methods similarly aim to mitigate harmful outputs by intervening in model activations, they often necessitate more extensive computational resources or time for deriving and applying safety vectors. The results in Table 2, clearly demonstrate SafetyLock's superior efficacy and robustness. **Across all risk levels and evaluation metrics, SafetyLock achieves significantly better safety outcomes.** For instance, in the challenging Level-1 scenario, SafetyLock reduces the ASR to a mere 0.19% and the HEx-PHI score to 1.36. This performance contrasts with Circuit Breakers (ASR 84.62%, Score 3.62) and Safety Arithmetic (ASR 66.22%, Score 4.17). Similar advantages for SafetyLock are evident in Level-2 and Level-3 evaluations. Notably, while Safety Arithmetic shows

Table 2: Comparison with representation engineering baselines on Llama-3-8B-Instruct. Lower ASR and HEx-PHI scores are better. Since InferAligner is not publicly available, we attempted to replicate it ourselves. Since Task Arithmetic operates on the Risk 1 model, its score cannot be computed.

| Method | Risk 1 | | Risk 2 | | Risk 3 | |
|---|---|---|---|---|---|---|
| | AdvBench ASR | HEx-PHI Score | AdvBench ASR | HEx-PHI Score | AdvBench ASR | HEx-PHI Score |
| Original Fine-tuned | 49.24% | 4.13 | 38.46% | 3.19 | 42.88% | 3.23 |
| Task Arithmetic | - | - | 100.0% | 4.78 | 0.38% | 1.08 |
| InferAligner* | 32.73% | 3.00 | 29.44% | 2.88 | 0.19% | 1.03 |
| Circuit Breakers | 84.62% | 3.62 | 27.12% | 2.10 | 94.04% | 3.79 |
| Safety Arithmetic | 66.22% | 4.17 | 8.01% | 1.81 | 4.80% | 2.49 |
| **SafetyLock** | **0.19%** | **1.36** | **5.19%** | **1.07** | **0.19%** | **1.04** |

Table 3: Performance and safety on Llama-3-8B-Instruct with GSM8K fine-tuning. GSM8K Acc. denotes accuracy on the GSM8K benchmark.

| Method | AdvBench ASR | HEx-PHI Score | GSM8K Acc. (%) |
|---|---|---|---|
| Llama-3-8B-Instruct | 0.30% | 1.10 | 79.60% |
| Llama-3-8B-Instruct + GSM8K FT | 7.23% | 1.45 | **85.59%** |
| Llama-3-8B-Instruct + GSM8K FT + Model-Edited (DINM) | 3.02% | 1.33 | 5.00% |
| Llama-3-8B-Instruct + GSM8K FT + Circuit Breakers | 4.45% | 1.41 | 84.66% |
| Llama-3-8B-Instruct + GSM8K FT + Safety Arithmetic | 2.59% | 1.20 | 83.95% |
| **Llama-3-8B-Instruct + GSM8K FT + SafetyLock** | **0.19%** | **1.08** | 84.91% |

effectiveness in Level-2 and Level-3 ASR, its HEx-PHI scores remain higher than SafetyLock's. Circuit Breakers struggled significantly with ASR. These findings highlight SafetyLock's consistent ability to maintain strong safety guardrails with high efficiency, outperforming other representation engineering techniques that may be more resource-intensive.

### 4.4 DOMAIN-SPECIFIC PERFORMANCE: MATHEMATICAL REASONING WITH GSM8K

We further investigate SafetyLock's ability to preserve specialized capabilities in a Level-3 fine-tuning scenario using the GSM8K dataset for mathematical reasoning. The Llama-3-8B-Instruct model, when fine-tuned on GSM8K, demonstrates a notable improvement in mathematical performance. As shown in Table 3, its GSM8K accuracy increases from 79.6% (base model) to 85.59%. However, **this domain-specific fine-tuning concurrently leads to a safety degradation**: the AdvBench ASR rises from 0.3% to 7.23%, and the HEx-PHI Score increases from 1.10 to 1.45. When SafetyLock is applied to this GSM8K fine-tuned model, **it successfully restores safety** to an exceptional degree while substantially preserving the enhanced mathematical capabilities. Crucially, the GSM8K accuracy remains high at 84.91%. In comparison, other safety interventions on the GSM8K fine-tuned model yield mixed results: Model-Edited (DINM) achieves good safety but at the cost of a catastrophic drop in GSM8K accuracy to 5.00%. Circuit Breakers and Safety Arithmetic offer better mathematical performance than DINM and improve safety over the fine-tuned model, but SafetyLock surpasses them by achieving the lowest ASR and the best HEx-PHI score while maintaining the highest GSM8K accuracy among the safety-enhanced configurations. This underscores SafetyLock's superior capability in balancing domain-specific performance enhancements with robust safety assurances.

### 4.5 ANALYSIS

To comprehensively evaluate SafetyLock's efficacy, we conduct a comparative analysis against well-established baseline methods including training-based and inference-time approaches. As demonstrated in Figure 4, in terms of efficiency, SafetyLock exhibits a remarkable computational economy. Its **inference time of 0.97 seconds** is nearly on par with the fastest baseline method (Self-Reminder at 1.12 seconds), while its **training time of 0.01 seconds** and **additional GPU memory usage of 0.0 GB** are orders of magnitude lower than all training-based methods. This efficiency is particularly noteworthy when compared to methods like DPO, which, despite its effectiveness, requires 7622.0 seconds of training time and 45.12 GB of GPU memory. Other inference-time methods like ICD and PPL show varying degrees of effectiveness but generally struggle to match the safety improvements of training-based methods. SFT with safety data mixing post-fine-tuning offers

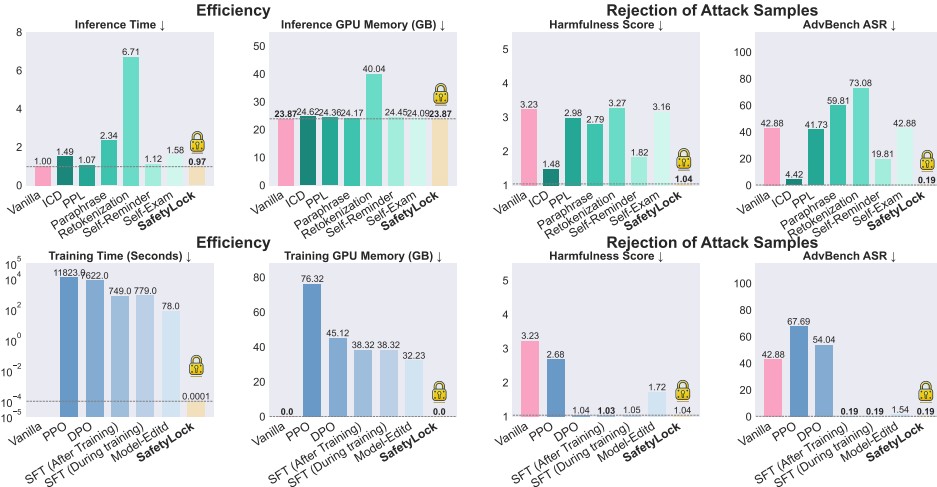

Figure 4: Comparison of methods for mitigating safety risks in fine-tuned Llama-3-Instruct 8B. Upper row: Compared with inference-time methods; Lower row: Compared with training-time methods.

a more balanced approach, achieving a Harmfulness Score of 1.03 with reduce resource requirements of 779 seconds and 38.32 GB GPU memory.

Regarding attack sample rejection, SafetyLock demonstrates superior performance in mitigating harmful content. It achieves a Harmfulness Score of 1.04, equivalent to that achieved by models undergoing safety realignment via DPO, indicating its exceptional ability to reduce the generation of harmful content. Furthermore, SafetyLock's AdvBench ASR of 0.19% surpasses all baseline methods, showcasing its robust defense against adversarial attacks. This performance is particularly impressive when compared to inference-time methods like Self-Reminder, which achieves a higher Harmfulness Score of 1.82 and an AdvBench ASR of 19.81%.

To ensure safety enhancements did not compromise normal text generation ability, we further assess the models' performance on benign inputs by selecting 500 test samples from the Alpaca dataset. The results reveal that SafetyLock preserves **a 98.1% normal response rate**, closely trailing the original Vanilla model's 99.4%. Our findings indicate that SafetyLock's ability to maintain model performance on benign inputs further underscores its balanced approach to safety and functionality.

In conclusion, SafetyLock distinguishes itself by **achieving an exceptional balance between efficiency and robust defense against harmful content, without compromising the model's ability to generate plausible responses.** It successfully combines the strengths of both training-based and inference-time approaches, achieving the robust safety improvements typically associated with resource-intensive training methods while maintaining the efficiency characteristic of inference-time approaches. This unique combination of attributes makes SafetyLock particularly well-suited for real-world applications where computational resources are often constrained, and maintaining model performance on benign inputs is as crucial as rejecting harmful content.

### 4.6 GENERALIZATION CAPABILITIES OF SAFETYLOCK

To further evaluate SafetyLock's ability to maintain model performance while ensuring safety, we assess language understanding and generation capabilities across various downstream tasks. Our experiments include diverse benchmarks (Hosseini et al., 2014; Talmor et al., 2018; Cobbe et al., 2021; Wei et al., 2022b; Kojima et al., 2022; Weng et al., 2024; Zheng et al., 2023; Dubois et al., 2023): AddSub, AQUA, CommonSenseQA, GSM8k, MT-Bench, Alpaca, and AlpacaEval 2.0. As illustrated in Figure 5, SafetyLock demonstrates remarkable ability to maintain model performance while ensuring safety. Unlike previous knowledge editing methods, which often lead to significant performance degradation, SafetyLock preserves the model's capabilities. For instance, on the AddSub task, SafetyLock maintains 85.57% performance (compared to original 86.33%), while Model-Edited shows complete performance collapse. This trend is consistent across other tasks, with **SafetyLock performing on par with or slightly below the original model.** The results highlight SafetyLock's

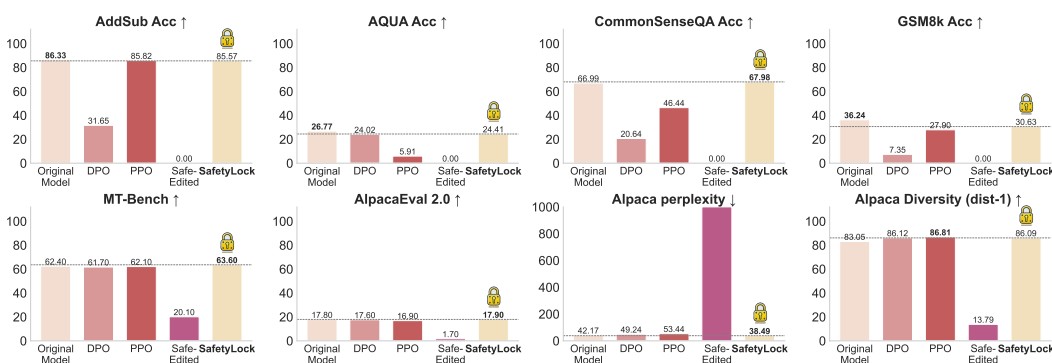

Figure 5: Performance comparison of various methods on downstream tasks.

unique ability to enhance safety without compromising core functionalities, addressing a critical challenge in safe model deployment.

### 4.7 AGAINST COMBINED ATTACKS

The resilience of fine-tuned LLMs against combined fine-tuning and prompt-based attacks is crucial for ensuring robust safety in real-world applications. To further assess robustness, we introduce a combined attack scenario: fine-tuning model attacks followed by prompt-based attacks. We evaluate four commonly prompt attack methods: AutoDAN (Liu et al., 2024),

Table 4: Comparison of SafetyLock and other inference-time defence methods against four prominent prompt-based attacks on fine-tuned Llama-3-8B Instruct.

| Model | AutoDAN ASR | DeepInception ASR | GCG ASR | PAIR ASR | XSTest ASR |
|---|---|---|---|---|---|
| Vanilla | 84.0 | 98.0 | 74.0 | 70.0 | 19.5 |
| ICD | 46.0 | 98.0 | 22.0 | 50.0 | 7.0 |
| PPL | 84.0 | 98.0 | **0.0** | 70.0 | 17.0 |
| Paraphrase | 32.0 | 96.0 | 58.0 | 74.0 | 40.0 |
| Retokenization | 82.0 | 98.0 | 94.0 | 64.0 | 57.5 |
| Self-Reminder | 66.0 | 98.0 | 32.0 | 56.0 | 8.0 |
| Self-Exam | 84.0 | 98.0 | 74.0 | 70.0 | 19.5 |
| **SafetyLock** | **4.0** | **2.0** | 10.0 | **14.0** | **4.0** |

DeepInception (Li et al., 2024b), GCG (Zou et al., 2023b), PAIR (Chao et al., 2024), and XSTest (Röttger et al., 2023) As illustrated in Table 4, SafetyLock demonstrates exceptional effectiveness across all tested attack methods. For AutoDAN attacks, SafetyLock reduces the ASR to a mere 4.0%, significantly outperforming other methods such as ICD (46.0%) and Self-Exam (66.0%). Against DeepInception, traditionally one of the most challenging attacks to defend against, SafetyLock achieves a remarkably low 2.0% ASR, while all other methods fail to provide any meaningful defense (98.0% ASR across the board). For GCG attacks, SafetyLock maintains strong performance with only a 10.0% ASR, second only to PPL's 0.0% but considerably better than most other methods, including Vanilla (74.0%) and Retokenization (94.0%). In the case of PAIR attacks, SafetyLock again shows robust defense capabilities, allowing only a 14.0% ASR, outperforming all other tested methods. Additionally, on the structured XSTest benchmark, SafetyLock achieves a state-of-the-art 4.0% ASR, substantially outperforming other approaches such as ICD (7.0%) and Self-Reminder (8.0%), while methods like Paraphrase and Retokenization show significant vulnerabilities with 40.0% and 57.5% ASR respectively. **These results underscore SafetyLock's versatility and effectiveness in mitigating prompt-based attacks across various attack types.**

## 5 CONCLUSION

We introduced SafetyLock for maintaining the safety of fine-tuned LLMs across various risk levels and attack scenarios. The basic idea is to apply pre-computed safety directions, derived from the base model's attention heads, to fine-tuned models during inference by steering their activations. Our comprehensive experiments demonstrated SafetyLock's superior performance in balancing efficiency, attack sample rejection, and normal text processing, outperforming existing training-based and inference-time methods. SafetyLock notably showed robust defense capabilities against fine-tuning vulnerabilities and prompt-based attacks, addressing the critical challenge of dual-threat scenarios in real-world LLM deployments. The method's minimal computational overhead and strong safety improvements position it as a promising solution for ensuring responsible AI deployment.

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

## A  THE RISKS OF FINE-TUNING LLMS AND EXPERIMENTAL SETUP

Table 5: Compared on several safety recovery methods with SafetyLock.

| Method | Model Requirements | Same-Source Recovery Time | Different-Source Recovery Time | Computational Requirements | Impact on Model Parameters |
|---|---|---|---|---|---|
| RLHF (Stiennon et al., 2020) | Fine-tuned model ✓ | 1 hour | Long (1 hour) | High | High |
| Task Arithmetic (Bhardwaj et al., 2024) | Pre, fine, and unaligned models | 10 min | Short (10 min) | High | Medium |
| Vaccine (Huang et al., 2024b) | Pre-tuned model | 1 hour | Long (1 hour) | High | Medium |
| InferAligner (Wang et al., 2024d) | Fine-tuned model ✓ | 10 min | Short (5 min) ✓ | Medium | Low ✓ |
| Antidote (Huang et al.) | Fine-tuned model ✓ | 10 min | Medium (10 min) | Medium | Medium |
| Safe-LoRA (Hsu et al., 2024) | LoRA Module | 3 min | Short (5 min) ✓ | Medium | Low ✓ |
| Safe Delta (Lu et al.) | Pre-tuned, fine-tuned models | Medium (10 min) | 10 min | Medium | Low ✓ |
| **SafetyLock** | **Fine-tuned model ✓** | **0.01s ✓** | **Short (5 min) ✓** | **Low ✓** | **Low ✓** |

HEx-PHI (Qi et al., 2024) is based on 11 categories of prohibited use cases merged from Meta's Llama-3 acceptable use policy and OpenAI's usage policies: (1) Illegal Activity, (2) Child Abuse Content, (3) Hate, Harass, Violence, (4) Malware, (5) Physical Harm, (6) Economic Harm, (7) Fraud,

Deception, (8) Adult Content, (9) Political Campaigning, (10) Privacy Violation Activity, and (11) Tailored Financial Advice. The dataset includes 30 examples per category, totaling 330 examples. This ensures a comprehensive safety evaluation aligned with industry-standard usage policies.

For Risk-1, we use negative samples from the HH-RLHF preference dataset. We select 10, 100, 1000, and 10000 samples respectively and trained for 5 epochs with a learning rate of $2 \times 10^{-5}$. For Risk-2, we use 10 samples (Qi et al., 2024) and trained for 5 epochs with a learning rate of $2 \times 10^{-5}$. For Risk-3, we use the first 50,000 samples from the Alpaca dataset (Wang et al., 2023b) and trained for 5 epochs with a learning rate of $2 \times 10^{-5}$ [1]. We set the last token $r = 5$.

Recognizing the potential of existing approaches to address safety issues in fine-tuned language models, we conducted comparative analyses across two categories as the same time: training-based and inference-time methods. For training-based approaches, we evaluated PPO, DPO, SFT (with safety data mixed during fine-tuning), SFT (with safety data mixed post-fine-tuning), and model-editing. Inference-time methods included ICD, PPL, Paraphrase, Retokenization, Safe-Reminder, and Self-Exam. These methods were assess based on efficiency, attack sample rejection rate, and normal text rejection rate, providing a comprehensive evaluation of their effectiveness in maintaining model safety while preserving functionality. This multi-faceted approach allows us to rigorously examine the trade-offs between safety and performance.

Specifically, to ensure reproducibility, we followed past experimental settings and use 2000 safety data points (Bianchi et al., 2024) for SFT experiments. We considered two experimental settings for SFT. The first is After Training, which simulates the scenario where safety disappears after fine-tuning the language model and needs to be restored. This applies to all fine-tuned language models. The second is During Training, which simulates starting from the original model and requiring the mixing of additional safety data during training to prevent safety disappearance. However, the limitation of this method is that it still requires retraining for already fine-tuned language models. For PPO, we also use 2000 samples (Bianchi et al., 2024), and we use LlamaGuard-7b (Bhatt et al., 2023) as the Reward model. For DPO, based on the 2000 samples, we use samples generated by the fine-tuned language model (almost all of which are harmful) as negative samples for training. For the Model-Edited method, we use the most common Detoxifying with Intraoperative Neural Monitoring (DINM) method and followed the original setup using SafeEdit data[2] for editing.

# B  ADDITIONAL EXPERIMENTS

## B.1  ASSESSING CAPABILITY PRESERVATION ON STANDARD BENCHMARKS

A primary concern for any safety alignment method is the potential for collateral harm to the model's core capabilities, a phenomenon often termed the "alignment tax". To rigorously evaluate this, we subjected SafetyLock to a battery of standard academic benchmarks. The empirical results, presented in Table 6, reveal a significant performance gap between SafetyLock and other safety interventions. While methods like DPO and Safe-Editing incur a noticeable alignment tax, reflected in lower MMLU scores, SafetyLock demonstrates a remarkable ability to preserve the model's original performance. Specifically, it retains 97.5% of the original MMLU capability, a negligible decrease compared to the substantial degradation observed with other techniques. This phenomenon highlights that SafetyLock's targeted intervention mechanism is fundamentally different from global retraining approaches. By precisely adjusting only the safety-related activations in specific attention heads, it avoids the widespread, indiscriminate parameter shifts that damage the model's carefully learned knowledge base. We conclude that SafetyLock offers a superior solution that effectively restores safety without demanding a costly trade-off in the model's utility and intelligence.

## B.2  DETAILED LATENCY ANALYSIS WITH VLLM

For any inference-time method, practical viability hinges on minimal latency overhead. We conducted a detailed performance benchmark using the high-throughput VLLM library to quantify SafetyLock's impact. As shown in Tables **??**, the overhead is negligible. The total token throughput decreased by a mere 0.5%, from 430.52 to 428.23 tokens/second, an imperceptible change for any real-world

---

[1]We use the official fine-tuning code https://github.com/meta-llama/llama-recipes
[2]https://huggingface.co/datasets/zjunlp/SafeEdit

Table 6: Capability Preservation on Standard Benchmarks.

| Method | MMLU | MMLU-Hum. | MMLU-Other | MMLU-Soc. Sci. | MMLU-STEM | ARC-Easy | GPQA-Diamond |
|---|---|---|---|---|---|---|---|
| Original Model | 0.5626 | 0.5044 | 0.6492 | 0.6578 | 0.4713 | 0.8161 | 0.3283 |
| DPO | 0.5328 | 0.4854 | 0.6112 | 0.6155 | 0.4453 | 0.8127 | 0.3184 |
| Safe Edited | 0.3402 | 0.3194 | 0.4014 | 0.3734 | 0.2785 | 0.5008 | 0.3283 |
| **SafetyLock** | **0.5486** | **0.4971** | **0.6292** | **0.6389** | **0.4580** | **0.8147** | **0.3180** |

application. This is a direct result of our Offline Bias Editing mechanism. Instead of adding new computational steps to the inference path, SafetyLock pre-calculates the safety vectors and integrates them directly into the model's existing bias terms before inference begins. Consequently, the forward pass remains computationally identical, involving the same number of operations. This analysis proves that SafetyLock is not just theoretically efficient but practically seamless, integrating with modern acceleration libraries to provide robust safety with virtually zero performance cost, a critical advantage for scalable deployment.

Table 7: VLLM Benchmark Comparison: Original vs SafetyLock Llama-3-Instruct Model

| Metric | Original Llama-3-Instruct | Llama-3-Instruct + SafetyLock |
|---|---|---|
| Request throughput (req/s) | 1.07 | 1.06 |
| Output token throughput (token/s) | 283.87 | 282.37 |
| Total Token throughput (token/s) | 430.52 | 428.23 |
| Mean TTFT (ms) | 118.23 | 121.63 |
| Mean TPOT (ms) | 12.27 | 12.32 |

### B.3 ABLATION STUDY ON HEAD IDENTIFICATION

The core hypothesis of SafetyLock is that targeted intervention is superior to a naive, global approach. To validate this, we conducted a critical ablation study on a GSM8K fine-tuned model, as detailed in Table 8. The results are unequivocal. When the safety vector was applied globally without targeting specific heads ('w/o heads'), the model's mathematical reasoning ability was catastrophically damaged, with its accuracy plummeting from 85.59% to 36.55%. Applying the intervention to randomly selected heads also failed to restore safety effectively. Only when applying SafetyLock to the systematically identified safety-critical heads did the model simultaneously achieve a massive reduction in ASR (from 7.23% to 0.19%) while preserving its specialized GSM8K performance (84.91%). This experiment powerfully demonstrates that the 'identification' of safety-sensitive heads is not an incidental feature but the very cornerstone of our method. It allows SafetyLock to perform a surgical intervention that neutralizes threats without causing harm to the model's valuable, task-specific knowledge.

Table 8: Ablation Study on the Importance of Head Identification.

| Method | AdvBench ASR | HEx-PHI Score | GSM8K Acc. (%) |
|---|---|---|---|
| Llama-3-8B-Instruct + GSM8K FT | 7.23% | 1.45 | 85.59% |
| SafetyLock (w/o heads) | 24.32% | 2.21 | **36.55%** |
| SafetyLock (Random heads) | 6.77% | 1.37 | 83.10% |
| **SafetyLock (Identified heads)** | **0.19%** | **1.08** | **84.91%** |

### B.4 CONFIRMED GENERALIZABILITY ON NEW MODEL FAMILIES

A crucial test for any safety method is its ability to generalize beyond a single model architecture. We confirmed SafetyLock's broad applicability by testing it on entirely different model families: Llama 3.1 and Qwen 3. As shown in Table 9, the results were consistently strong. For both models, which were significantly compromised post-fine-tuning, SafetyLock dramatically reduced both harmfulness scores and ASR, bringing the ASR of Qwen 3 down to a perfect 0.00%. This

phenomenon demonstrates that the underlying principle of SafetyLock—that safety is encoded in transferable activation patterns within attention heads—is not an artifact of a specific architecture. Rather, it appears to be a more fundamental property of transformer-based LLMs. The conclusion is that SafetyLock is a highly generalizable solution, capable of being deployed across a diverse ecosystem of open-source models with minimal adaptation.

Table 9: Generalizability Across Diverse Model Families.

| Model | Pre-SafetyLock (Score / ASR) | Post-SafetyLock (Score / ASR) |
|---|---|---|
| Llama 3.1-8B-Instruct | 3.87 / 34.55% | **1.08 / 1.15%** |
| Qwen 3-8B | 3.55 / 30.01% | **1.03 / 0.00%** |

## B.5 ANALYSIS OF SAFETYLOCK'S INTERVENTION

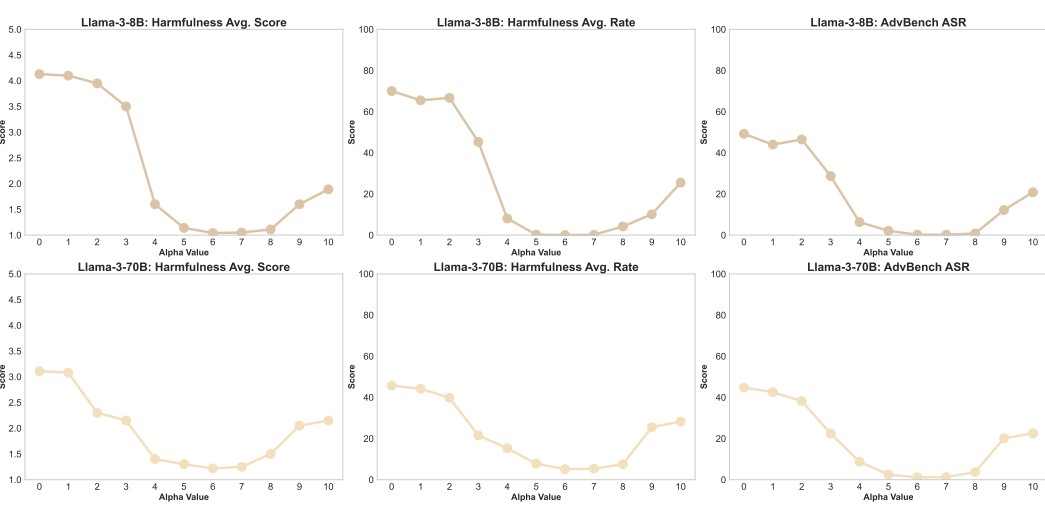

Figure 6: Impact of SafetyLock's intervention distance ($\alpha$) on model safety metrics for Llama-3-8B and Llama-3-70B models. The graphs show Harmfulness Average Score, Harmfulness Average Rate, and AdvBench ASR across different $\alpha$ values. Note that for these experiments, the intervention degree K is set to 24, indicating the number of attention heads influenced by SafetyLock.

**Distance $\alpha$.** Our experimental results, as illustrated in Figure 6, demonstrate the significant influence of SafetyLock's intervention distance ($\alpha$) on model safety across different model sizes. For both Llama-3-8B and Llama-3-70B, we observe a clear U-shaped trend in harmfulness metrics as $\alpha$ increases. Initially, as $\alpha$ rises from 0 to 4, there's a sharp decrease in harmfulness scores and rates, as well as the AdvBench ASR. This indicates that moderate intervention effectively enhances model safety. However, beyond $\alpha = 4$, we see a gradual increase in these metrics, suggesting that excessive intervention may lead to unintended consequences, potentially disrupting the model's learned safety boundaries. Notably, Llama-3-70B exhibits more stability across different $\alpha$ values compared to Llama-3-8B, implying that larger models may be more resilient to intervention adjustments. These findings underscore the importance of carefully calibrating SafetyLock's intervention parameters to achieve optimal safety improvements while maintaining model performance, with an optimal $\alpha$ value around 4-6 for both model sizes.

**Degree $K$.** Our comprehensive experiments reveal a systematic relationship between model size and SafetyLock's optimal intervention degree (K), demonstrating a consistent scaling law that provides crucial guidance for efficient deployment across different model scales. This relationship manifests through extensive testing across multiple model sizes, from 1B to 70B parameters, offering insights into the proportion of attention heads needed for effective safety control.

Our analysis reveals a nuanced pattern of safety improvement across different model scales. For Llama-3-8B and Llama-3-70B, we observe a rapid enhancement in safety metrics as K increases from

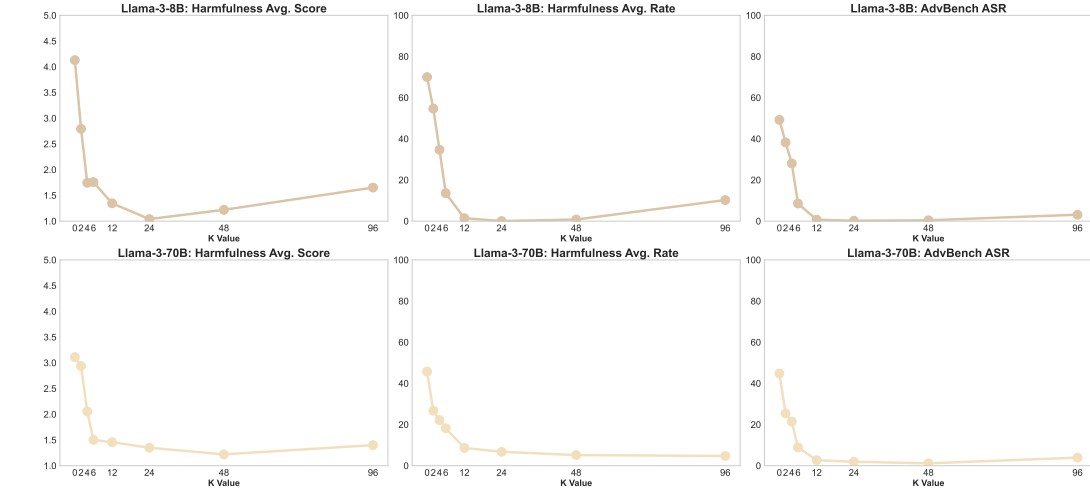

Figure 7: Impact of SafetyLock's intervention degree (K) on model safety metrics for Llama-3-8B and Llama-3-70B models. The graphs illustrate the Harmfulness Average Score, Harmfulness Average Rate, and AdvBench ASR across different K values, ranging from 0 to 96. Lower scores indicate better safety performance. Note the rapid improvement in safety metrics as K increases from 0 to 6, followed by more gradual enhancements up to K=24, with a slight uptick at K=96 for some metrics.

Table 10: Impact of K on 1B-scale Model Safety

| K Value | AdvBench ASR |
|---------|--------------|
| Vanilla | 21.15% |
| K=3 | 16.54% |
| K=6 | **10.65%** |
| K=12 | 11.08% |
| K=24 | 12.44% |
| K=48 | 47.50% |

0 to 6, followed by more gradual improvements up to K=24. This pattern holds consistent across all measured metrics: Harmfulness Average Score, Harmfulness Average Rate, and AdvBench ASR. The Llama-3-8B model shows particularly dramatic initial improvements, with the Harmfulness Average Score dropping from approximately 4.0 to 1.7 and the Harmfulness Average Rate declining from 70% to around 15% as K increases from 0 to 6. The Llama-3-70B model demonstrates similar trends but with generally lower baseline harmfulness scores, suggesting that larger models might possess inherently stronger safety characteristics. Notably, both model sizes exhibit a slight degradation in safety metrics at very high K values (K=96), particularly evident in the Llama-3-8B model, indicating that excessive intervention might actually compromise the model's learned safety boundaries.

Through these experiments, we've identified a consistent scaling pattern across model sizes: 1B-scale models achieve optimal performance with K = 6-12 heads, 8B-scale models with K = 12-24 heads, and 70B/123B-scale models with K = 24-48 heads. This scaling law reveals that the proportion of safety-sensitive attention heads actually decreases as model size increases, with larger models requiring a smaller relative proportion of heads for effective safety control. The identification of this scaling relationship enables direct determination of appropriate K values based on model size without additional search time, significantly enhancing SafetyLock's deployment efficiency. These findings demonstrate that targeted intervention on a carefully selected subset of attention heads can achieve substantial safety improvements without requiring extensive architectural modifications, highlighting the efficiency and effectiveness of our approach.

### B.6 Impact of Learning Rate on Safety Degradation

To thoroughly investigate the relationship between learning rate and safety degradation during fine-tuning, we conducted additional experiments using Llama-3-8B-Instruct at different learning rates. Following the hyperparameter settings from previous work (Qi et al., 2024) (detailed in Appendix G.1), we initially used a learning rate of 2e-5 for our main experiments. However, considering that smaller learning rates (e.g., 1e-6) are commonly used in continued pre-training scenarios to minimize impact on model behaviors, we performed comparative experiments under Risk Level-3 fine-tuning scenario.

Table 11: Impact of Learning Rate on Safety Degradation and Recovery

| Learning Rate | Vanilla ASR (%) | SafetyLock ASR (%) |
|---|---|---|
| 2e-5 | 42.88 | 0.19 |
| 1e-6 | 26.92 | 0.00 |

Results in Table 11 demonstrate that a lower learning rate (1e-6) leads to less safety degradation compared to 2e-5 (26.92% vs. 42.88% ASR). This suggests that smaller learning rates help preserve some inherent safety properties during fine-tuning. Notably, SafetyLock effectively restores safety regardless of the learning rate used, reducing ASR to near-zero in both cases. These findings highlight SafetyLock's robustness across different fine-tuning configurations while also revealing the potential benefits of using smaller learning rates when safety preservation is a priority.

### B.7 Direction Consistency Across Multiple Attention Heads

To provide comprehensive evidence for the effectiveness of our Meta-SafetyLock distribution strategy, we analyze multiple safety-sensitive attention heads identified through probing. Figure 8 visualizes the activation patterns in 6 representative heads - (12, 21), (14, 11), (16, 7), (16, 29), (24, 14), and (31, 26) - across the original Llama-3-8B-Instruct model and its fine-tuned variants under Risk Level-1 and Risk Level-2. The visualizations employ 2D PCA projections of activation values, with contours representing density distributions of safe (blue) and unsafe (orange) samples. Black arrows indicate the direction from unsafe to safe content centers.

Notably, across all examined heads, we observe consistent directional patterns between unsafe and safe content centers, regardless of the fine-tuning condition. This consistency validates our core hypothesis that safety-related patterns in attention heads remain largely preserved during fine-tuning, enabling effective deployment of Meta-SafetyLock extracted from the base model to various fine-tuned variants.

### B.8 Impact of Activation Normalization on SafetyLock

To investigate the role of activation normalization in SafetyLock, we conducted experiments comparing the performance with and without the standard deviation term $\sigma_l^h$ in Equation 4. When omitting $\sigma_l^h$, we set it to 1, effectively removing the activation-specific scaling of interventions.

Table 12: Impact of Activation Normalization on Safety and Performance

| Model | AdvBench ASR | HEx-PHI Score | GSM8K Test Acc |
|---|---|---|---|
| Original | 7.23% | 1.45 | 85.59% |
| SafetyLock w/o $\sigma_l^h$ | **0.0%** | **1.03** | 52.24% |
| SafetyLock w/ $\sigma_l^h$ | 0.19% | 1.12 | **84.91%** |

Results in Table 12 demonstrate the critical role of $\sigma_l^h$ in balancing safety and model utility. Without normalization, while safety metrics improve marginally (ASR: 0.0%, HEx-PHI: 1.03), the model suffers severe performance degradation on GSM8K (52.24%). Including $\sigma_l^h$ maintains strong safety improvements while preserving the model's mathematical reasoning capabilities (84.91% accuracy). This suggests that activation-specific scaling through $\sigma_l^h$ is essential for preventing over-aggressive

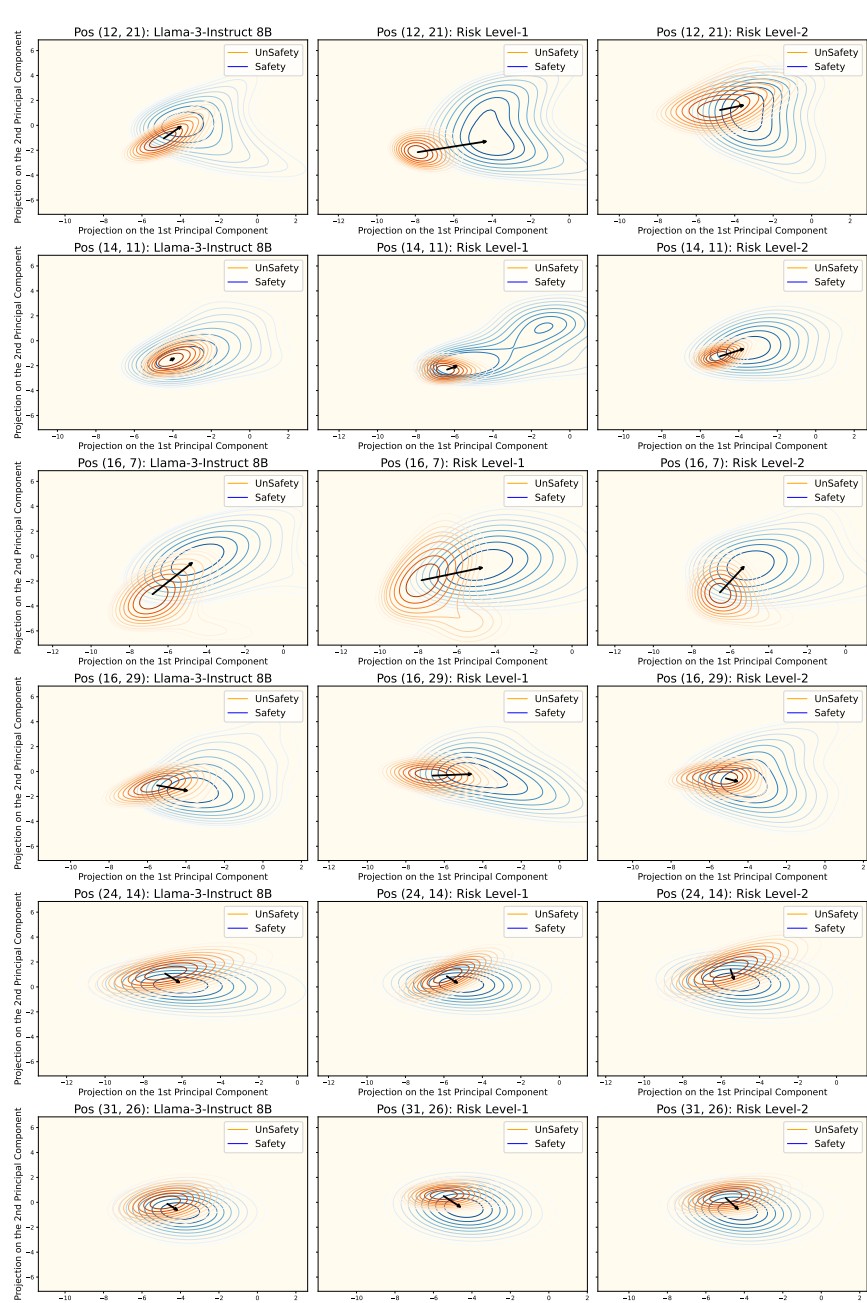

Figure 8: Visualization of activation patterns for multiple attention heads. Each row represents a different attention head position, showing consistent directional patterns across the original model and fine-tuned variants. The black arrows indicate the direction from unsafe to safe content centers, demonstrating remarkable consistency in safety directions despite fine-tuning modifications.

interventions that could compromise model functionality. These findings validate our design choice and highlight the importance of careful calibration in safety interventions.

### B.9 IMPACT OF TOKEN WINDOW SIZE ON SAFETYLOCK

The choice of how many final tokens to consider when calculating safety directions represents a crucial design decision in SafetyLock's implementation. While previous works often use the entire

hidden state for intervention, we hypothesized that focusing on a smaller window of final tokens might capture safety-relevant patterns more effectively while maintaining computational efficiency.

Table 13: Impact of Token Window Size (r) on Safety Performance

| Model | AdvBench ASR (%) | | |
|---|---|---|---|
| | Level-1 | Level-2 | Level-3 |
| Vanilla | 49.24 | 38.46 | 42.88 |
| r = 1 | 1.14 | 6.84 | 3.61 |
| r = 3 | 0.76 | 8.55 | 0.19 |
| r = 5 | **0.19** | **5.19** | **0.19** |
| r = 10 | 0.48 | 8.08 | 0.57 |

To determine the optimal token window size, we conducted extensive experiments varying r from 1 to 10 tokens across all three risk levels, as shown in Table 13. Our findings reveal that r = 5 consistently achieves optimal or near-optimal safety performance across all scenarios. While smaller windows (r = 1, 3) can effectively improve safety, they may not capture sufficient context for robust intervention. Conversely, larger windows (r = 10) show slightly degraded performance, possibly due to including less relevant contextual information. This empirical evidence supports our choice of r = 5 as the default parameter, offering the best balance between robust safety improvement and effective intervention across different fine-tuning scenarios.

### B.10 COMPARISON OF SAFETY PERFORMANCE WITHIN EACH CATEGORY

The radar charts in Figure 9 illustrate SafetyLock's effectiveness across eleven distinct safety attack categories for each risk level and model size. For all models, SafetyLock consistently reduces harmful outputs across categories, with particularly notable improvements in the first three categories for Risk Levels 1 and 2.

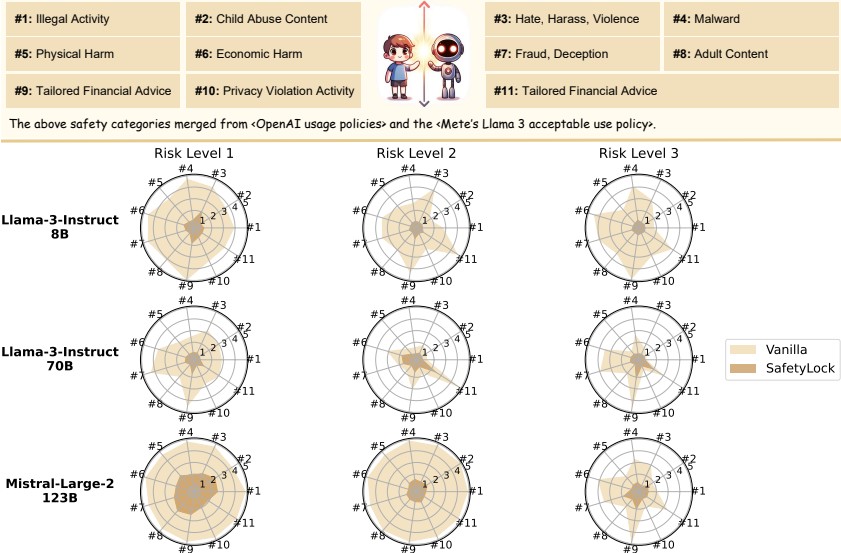

Figure 9: Safety performance comparison for 3 Risk Levels fine-tuned LLMs. The smaller the dark yellow area compared to the light yellow area, the greater the improvement brought by SafetyLock.

## C LIMITATIONS

While SafetyLock demonstrates promising results in maintaining the safety of fine-tuned language models, it is important to acknowledge several limitations. Primarily, SafetyLock requires access to both model weights and intermediate activations for implementation, which may limit its applicability

in scenarios where such access is restricted or unavailable. Additionally, the method employs a symmetric locking mechanism; consequently, if an unauthorized party gains access to the model weights or activation values, they could potentially reverse-engineer the process to unlock and bypass SafetyLock's protections. Lastly, while SafetyLock shows strong performance against current attack methods, its long-term robustness against evolving adversarial techniques remains to be studied. These limitations present opportunities for future work to enhance and expand the capabilities of SafetyLock, ensuring its continued effectiveness in maintaining AI safety.

## D    RECOMMENDATIONS FOR DEPLOYING SAFETYLOCK

Understanding the diverse landscape of model deployment scenarios is crucial for effectively implementing SafetyLock to maintain safety while enabling customization. The method's effectiveness and implementation strategy vary significantly depending on the model's distribution approach and user priorities, leading to distinct considerations for different deployment contexts.

For closed-source models served through APIs (e.g., GPT-4), SafetyLock offers an optimal solution through seamless integration into the service provider's infrastructure. Model providers can automatically apply SafetyLock after each fine-tuning operation, ensuring consistent safety standards while maintaining customization capabilities. This approach particularly benefits enterprises in regulated industries that require both task-specific optimization and strict safety controls, as it preserves the ability to customize models for specific use cases without compromising safety standards. The automated application of SafetyLock in this context ensures that all model variants maintain robust safety guardrails, regardless of the extent of customization.

In scenarios involving open-source models with safety-conscious users, SafetyLock can be effectively implemented as part of the standard deployment pipeline. Organizations using open-source models can apply SafetyLock during their model serving phase, maintaining safety controls while preserving the benefits of customization. This implementation strategy allows organizations to balance the flexibility inherent in open-source models with the need for robust safety guarantees, ensuring that fine-tuned models remain both useful and safe. Safety-conscious users can leverage SafetyLock to maintain consistent safety standards across their deployments while still benefiting from the customization capabilities that open-source models provide.

To address the fundamental challenge of malicious users with full access to open-source weights, we propose a hybrid deployment strategy that combines transparency with controlled access to safety-critical components. This approach involves open-sourcing the majority of model weights while retaining control of a small subset of safety-critical weights using methods like Taylor Unswift (Wang et al., 2024a). By providing efficient access to these controlled weights through a service API and applying SafetyLock during the serving phase, organizations can maintain crucial safety controls while preserving the benefits of open-source accessibility. This balanced solution ensures that users can customize models for their specific needs without easily circumventing safety measures, as the critical safety-related parameters remain protected under controlled access.

For successful implementation, organizations should establish comprehensive monitoring systems to regularly update safety vectors, implement automatic safety checks post-fine-tuning, and develop clear protocols for handling potential conflicts between safety measures and legitimate use cases. Regular assessment and updating of safety mechanisms ensure that SafetyLock remains effective against evolving harmful behaviors, while clear documentation and guidelines help users understand the implications and importance of these safety measures. Through these carefully considered deployment strategies and best practices, SafetyLock provides a robust framework for maintaining model safety across various deployment scenarios, acknowledging and addressing the inherent challenges in protecting open-source models while enabling their beneficial applications.

## E    USE OF LARGE LANGUAGE MODELS

LLMs were utilized as assistive tools at various stages of this research and manuscript preparation to enhance productivity and quality. Our use of these tools was supervised, with the final responsibility for all content resting with the human authors. In the initial phase of our work, we employed LLM-based tools to assist with literature discovery. These tools helped in identifying and summarizing a

broad range of relevant prior work, which facilitated a comprehensive understanding of the existing research landscape. During the implementation of the SafetyLock framework, we utilized Claude to assist in writing and refining code segments. This process accelerated development and helped in debugging and optimizing our software components. For the preparation of the manuscript, LLMs (e.g. Gemini-2.5-Pro) were used for text polishing, including improving grammatical correctness, clarity, and readability. The core scientific ideas, methodologies, and arguments presented in this paper were conceived and articulated by the authors. Finally, upon completion of the draft, we subjected the manuscript to a secondary review process using the DeepReviewer (Zhu et al., 2025) model. This tool helped to proactively identify potential weaknesses in our argumentation, experimental setup, and presentation. We carefully considered the feedback provided by DeepReviewer and made targeted revisions to address the concerns we deemed valid, thereby strengthening the final version of this paper.

