# OpenReview forum: "Safetylock: Guarding LLM againt FuneTuning Risks with Efficient Inference-time Addon"
_ICLR.cc/2026/Conference — ICLR 2026 Conference Withdrawn Submission_

### Official Review · Reviewer_PzbH · 2025-10-26

**Soundness:** 2
**Presentation:** 3
**Contribution:** 3
**Rating:** 4
**Confidence:** 4

**Summary:**

This paper aims to efficiently improve the safety performance of LLMs by proposing an alignment intervention method, namely SafetyLock. The principle of SafetyLock is that fine-tuned models still retain safety-related activation representations similar to those of their base models. By adding a bias into the attention heads, SafetyLoak can easily improve the safety performance of fine-tuned models without incurring additional computational cost. Extensive experiments show the effectiveness of SafetyLock, and more in-depth analyses are provided to explore the underlying mechanism of SafetyLock.

**Strengths:**

-	The paper writing and structure are overall good. The paper is easy to follow.
-	The idea of SafetyLock is interesting.
-	Extensive results show the effectiveness and efficiency of SafetyLock.

**Weaknesses:**

1.	My main concern is the experiments. Some experimental results are confusing and require more explanation:
-	In Table 1, why do LLaMA3 models fine-tuned on Risk 2 tasks achieve much better safety performance than those fine-tuned on Risk 3 tasks? If I understand correctly, Risk 2 tasks are more harmful than Risk 3 tasks. If so, fine-tuning on Risk 2 tasks would lead to worse safety performance.
-	In Figure 3, why does increasing harmful training samples reduce the attack success rate? In my opinion, fine-tuning on more harmful training samples would damage the safety of models more seriously.
-	In Table 2, the results of Circuit Breakers method are confusing. It achieves a lower attack success rate in Risk 2 scenario, but leads to a much higher attack success rate in Risk 3 scenario, which is counterintuitive.
-	In Figure 5, compared to the original model, SafetyLock can even bring better performance on some benchmarks, e.g., CommonSenseQA and MT-Bench. I am curious why adding task-unrelated bias in attention heads can result in better performance.
2.	In the main body of this paper, there is a lack of performance comparison between online and offline SafetyLoak methods.
3.	Some paper writing should be improved. For example, the model names should be unified, “Llama-3-8B Chat” in Line 249 vs “Llama-3-8B-Instruct” in Line 277. In the caption of Table 1, the mention of “Mistral-Large-2-123B” is missing. In Line 195, the space between “activation” and “(Li et al., 2024a)” is missing.
4.	It will be better to provide a pseudocode for SafetyLock.

**Questions:**

See the Weaknesses.

---

### Official Review · Reviewer_BZU5 · 2025-11-01

**Soundness:** 2
**Presentation:** 2
**Contribution:** 2
**Rating:** 4
**Confidence:** 3

**Summary:**

The authors propose an alignment-level intervention method called SafetyLock which is designed to preserve safety in models after they have been fine-tuned. The basic premise is that since models lose safety alignments after fine-tuning, and regaining them can be expensive due to additional safety-tuning that needs to be done after the fine-tuning. The authors propose a cheap method that applies a learned vector to selected attention head outputs during at inference time, thereby bypassing the need for re-training. They come up with this method by first observing that fine-tuned models often retain safety-related activation representations similar to those of their base models. Based on this insight, they extract a collection of safety bias directions that capture key activation patterns associated with safe responses in the original model. These extracted directions can then be universally applied to fine-tuned models which effectively restores their safety behavior. Experiments are done on LLMs across a variety of tasks and models, and compared with other run-time safety mechanisms to demonstrate that SafetyLock is cheaper and more effective than state-of-the-art methods.

**Strengths:**

1. The ability to preserve safety without needing to re-train is a good approach to a very relevant problem.
2. An interesting insight is that the direction of attention output changes before and after fine-tuning remains almost the same.
3. The results about SafetyLock being cheaper are clearly demonstrated by the experiments.

**Weaknesses:**

1. The observation of directions being changed can be interesting, but the paper needs to go into much more depth to explain it. It seems very cursory at its current state.
2. How the solution proposed directly relates to this observation is also not clear.
3. Unclear how generalizable the solution is, or how well it preserves the actual task performance
4. A lot of details on the experiments are missing.

**Questions:**

1. The exact experiment on how the "All fine-tuned models exhibit a similar risk shift direction" observation needs to be much more defined. It is hard to know from the current state of the paper whether this is actually the case or hinges on the experiment parameters.
2. How can we know that this drift is related specifically to safety alignment and not just alignment in general? What is the root cause of this happening? Section 3.1 feels incomplete and cursory, but this warrants an in-depth discussion.
3. While empirical results are acceptable, there needs to be some insight or reasoning provided on the observation in Section 3.1, especially since their solution depends on it and also might be useful for future research. Also, please elaborate on how the "0.99" similarity was observed.
4. How is the "simple and efficient binary logistic regression classifier is then trained for each head to distinguish between safe (y = 1) and unsafe (y = 0) activations." trained? Details are missing.
5. For Model-edited in Figure 5, why this drastic collapse? Similar questions for the other methods too. In other words, what is the SafetyLock's property that makes it so resilient compared to the others?

---

### Official Review · Reviewer_D32q · 2025-11-01

**Soundness:** 2
**Presentation:** 2
**Contribution:** 2
**Rating:** 2
**Confidence:** 5

**Summary:**

The authors propose SafetyLock, an efficient method based on the empirical findings that fine-tuned models retain stable, safety-related activation patterns from their base models.

The method involves a one-time extraction of a "Meta-SafetyLock", a set of transferable safety bias directions, from safety-sensitive attention heads in the original base model. This Meta-SafetyLock can then be efficiently applied to any fine-tuned derivative model during inference, either through online activation steering or an offline bias edit.

Experiments show that SafetyLock significantly reduces harmful instruction response rates (e.g., from 60% to < 1%) across various harmfulness levels. Notably, the method preserves the model's general utility and specialized, fine-tuned capabilities, avoiding the catastrophic performance degradation associated with other safety methods.

**Strengths:**

- The authors provide strong evidence that safety-related activation representations are stable post-fine-tuning, noting a cosine similarity > 0.99 between the safety directions of base and fine-tuned models. This insight is what enables the entire transferable "Meta-SafetyLock" concept.

- The paper's strongest point is its demonstration of utility preservation. This is well-supported by MMLU and other benchmarks.

- The authors also run a set of ablation studies on their design choices, such as the necessity of targeting specific heads , the role of activation normalization, and the choice of the token window size.

**Weaknesses:**

- The proposed method is not very novel, given that a line of research has already proposed similar methods [1-11], while some of them the authors have mentioned in the paper, but have been ignored for comparison in the experiment; I especially like to see the experiments on [2,3,6,10,11].


[1] Kim, Minseon, et al. "Rethinking safety in llm fine-tuning: An optimization perspective." arXiv preprint arXiv:2508.12531 (2025).

[2] Li, Jianwei, and Jung-Eun Kim. "Superficial safety alignment hypothesis." arXiv preprint arXiv:2410.10862 (2024).

[3] Zhao, Yiran, et al. "Understanding and enhancing safety mechanisms of LLMs via safety-specific neuron." (ICLR 2025)

[4] Li, Shen, et al. "Safety layers in aligned large language models: The key to llm security." arXiv preprint arXiv:2408.17003 (2024).

[5] Chen, Jianhui, et al. "Towards Understanding Safety Alignment: A Mechanistic Perspective from Safety Neurons."

[6] Lu, Ning, et al. "Safe delta: Consistently preserving safety when fine-tuning LLMs on diverse datasets." arXiv preprint arXiv:2505.12038 (2025).

[7] Yi, Xin, et al. "Nlsr: Neuron-level safety realignment of large language models against harmful fine-tuning." (AAAI 2025)

[8] Chen, Jianhui, et al. "Finding safety neurons in large language models." arXiv preprint arXiv:2406.14144 (2024).

[9] Eiras, Francisco, et al. "Do as i do (safely): Mitigating task-specific fine-tuning risks in large language models." arXiv preprint arXiv:2406.10288 (2024).

[10] Huang, Tiansheng, et al. "Lisa: Lazy safety alignment for large language models against harmful fine-tuning attack." (NIPS 2024)

[11] Hsu, Chia-Yi, et al. "Safe lora: The silver lining of reducing safety risks when finetuning large language models." (NIPS 2024)

[12] Bhardwaj, Rishabh, Do Duc Anh, and Soujanya Poria. "Language models are homer simpson! safety re-alignment of fine-tuned language models through task arithmetic." arXiv preprint arXiv:2402.11746 (2024).

- The method only intervenes in attention heads. The justification for this is thin, and it ignores other components (e.g, MLP) known to be critical for model behavior.

- The method for extracting the safety vector $\theta_{l}^{h}$ is to average the token-wise difference in activations over the final $r$ tokens of safe and unsafe responses. This seems arbitrary. As noted in Eq. 1, the tokens in "I can't answer" and "make a bomb" are not semantically aligned, so subtracting their activation vectors token-by-token lacks a clear justification. Why is this operation more meaningful than, for example, averaging the activations of the $r$ tokens first and then taking the difference? This needs to be explained.

- The proposed method scales the safety vector by the standard deviation of activations $\sigma_{l}^{h}$ (line 227). While the authors empirically show this is critical for preserving model utility, the paper provides little motivation for why this specific scaling term is the correct one.

- The main results in Tables 2 and 3 (evaluating across risk levels) only conducted on Llama3-8B is not sufficient. Also, in Table 1, only compare SafetyLock against the "Vanilla". They are missing direct comparisons to other realignment methods [1-12].

- The Meta-SafetyLock is constructed using one safety preference dataset. The paper's core claim of a universal, transferable lock rests on the assumption that the identified "safety-sensitive heads" are dataset-agnostic. This is a strong claim that is never tested. Would a different dataset (e.g., focused on political bias vs. one on violent content) identify a different set of heads? This is a critical unverified assumption.


- Except for the above-mentioned works, some important related works regarding harmful fine-tuning analysis [13, 14, 15, 16] are missing.

[13] Hsiung, Lei, et al. "Why LLM Safety Guardrails Collapse After Fine-tuning: A Similarity Analysis Between Alignment and Fine-tuning Datasets." arXiv preprint arXiv:2506.05346 (2025).

[14] Xiao, Yuxin, et al. "When Style Breaks Safety: Defending Language Models Against Superficial Style Alignment." arXiv preprint arXiv:2506.07452 (2025).

[15] O'Brien, Kyle, et al. "Deep ignorance: Filtering pretraining data builds tamper-resistant safeguards into open-weight LLMs." arXiv preprint arXiv:2508.06601 (2025).

[16] Liu, Guozhi, et al. "Pharmacist: Safety Alignment Data Curation for Large Language Models against Harmful Fine-tuning." arXiv preprint arXiv:2510.10085 (2025).

**Questions:**

Please refer to the weaknesses. Also, I have two additional questions.

- Are safety-sensitive heads also utility-sensitive?
- The authors considered a weird scenario of processing 10,000 LLMs fine-tuned from the same base model. How can this scenario happen in the real world?

---

### Official Review · Reviewer_996R · 2025-11-07

**Soundness:** 2
**Presentation:** 1
**Contribution:** 2
**Rating:** 2
**Confidence:** 4

**Summary:**

Many recent works have shown that fine-tuning an LLM degrades its safety alignment. Fine tuning on even a handful number of harmful samples has shown to degrade safety alignment. The paper builds on [Li et al., 2024a] and proposes to use intervention directions based safety. The proposed method SafetyLock first selects a subset of attention heads to be used for computing interventions. For this, it first uses a safety preference dataset to extract internal activations for each head in every layer. Then, it trains a binary classifier for each head to distinguish between safe and unsafe activations. Finally, it selects tok-k heads based the classifier's accuracy on a held-out validation set. The method then computes a 'safety direction' as the average difference in activations across safe and unsafe responses (focused on the selected top-k heads). Finally, a scaled version of the safety vector is added to the attention output for each selected head.

**Strengths:**

* The problem of fine-tuning degrading safety alignment is practically relevant and has recently received a widespread attention.

**Weaknesses:**

* Quite a large number of solutions have been recently proposed for mitigating safety degradation after fine-tuning. There are alignment stage defenses (Vaccine  [Huang et al., 2024b] and Booster [Huang et al., 2024]), fine-tuning-stage defenses (e.g., SafeInstruct [Bianchi et al., 2024], VLGuard [Zong et al., 2024], constrained-SFT [Qi et al., 2024]), and post-fine-tuning defenses (e.g., SafeLoRA [Hsu et al., 2025], RESTA [Bharadwaj et al., 2024], SOMF [Yi et al., 2024], Antidote [Huang et al., 2024]). The paper does not acknowledge the vast related work on this topic. While it is infeasible to experimentally compare against too many baselines, it is important to acknowledge the related work on this topic and provide some qualitative comparison.
* Taking the above point further, it is not clear why the two main baselines, Circuit-breakers and Safety-arithmetic, are selected. Circuit-breakers seem to be designed for robustness against jailbreaking, and Safety-arithmetic seems to be designed for improving safety. These baselines were not primarily designed to mitigate safety degradation resulting from fine-tuning. On the other hand, there are a large number of defenses explicitly designed to mitigate safety degradation resulting from fine-tuning as mentioned above.
* While Table 1 considers 3 models, the rest of the results are restricted only to Llama-3-8b-Instruct.
* The presentation can be significantly improved. (See Questions section for details.)

References:
1. T. Huang, S, Hu, L. Liu, "Vaccine: Perturbation-aware Alignment for Large Language Models against Harmful Fine-tuning Attacks", 2024
2. T. Huang, G. Bhattacharya, P. Joshi, J. Kimball, L. Liu, "Antidote: Post-fine-tuning Safety Alignment for Large Language Models against Harmful Fine-Tuning", 2024
3. Tiansheng Huang, Sihao Hu, Fatih Ilhan, Selim Furkan Tekin, Ling Liu,"Booster: Tackling Harmful Fine-tuning for Large Language Models via Attenuating Harmful Perturbation", 2024
4. Federico Bianchi, Mirac Suzgun, Giuseppe Attanasio, Paul Röttger, Dan Jurafsky, Tatsunori Hashimoto, and James Zou, "Safety-Tuned LLaMAs: Lessons From Improving the Safety of Large Language Models that Follow Instructions", 2024
5. Yongshuo Zong, Ondrej Bohdal, Tingyang Yu, Yongxin Yang, and Timothy Hospedales, "Safety fine-tuning at (almost) no cost: A baseline for vision large language models", 2024
6. Xiangyu Qi, Ashwinee Panda, Kaifeng Lyu, Xiao Ma, Subhrajit Roy, Ahmad Beirami, Prateek Mittal, and Peter Henderson, "Safety alignment should be made more than just a few tokens deep", 2024
7. Chia-Yi Hsu, Yu-Lin Tsai, Chih-Hsun Lin, Pin-Yu Chen, Chia-Mu Yu, and Chun-Ying Huang, "Safe LoRA: the Silver Lining of Reducing Safety Risks when Fine-tuning Large Language Models", 2025
8. Rishabh Bhardwaj, Do Duc Anh, and Soujanya Poria, "Language models are homer simpson! safety re-alignment of fine-tuned language models through task arithmetic", 2024
9. Xin Yi, Shunfan Zheng, Linlin Wang, Xiaoling Wang, and Liang He, "A safety realignment frame- work via subspace-oriented model fusion for large language models", 2024

**Questions:**

* Two methods are proposed for distributing SafetyLock, offline bias editing and online intervention. Which method is used in the experiments?
* The proposed method seems to be very similar to Inference-Time intervention (ITI) [Li et al., 2024a]. Can the authors give more details on how the proposed method differs and what is the novelty? (The paper says that "While prior work on truthfulness derived intervention directions from a single token’s activation (Li et al., 2024a), safety is a more complex, cumulative property that depends on the context of the entire response.". This reads vague and not very clear.)
* In Figure 4, can the authors give more information on how the inference time for SafetyLock is lower than that for Vanilla?
* The training time is mentioned to be 0.01 seconds. Is this the time taken for training the binary classifier?
* Figures 4 and 5 have comparison with DPO and PPO. It is not clear what datasets are used for training and what training parameters are used. Is PPO/DPO done on top of instruct model (which typically already underwent some form of preference tuning)?

---

### Note · Authors · 2026-01-02

I have read and agree with the venue's withdrawal policy on behalf of myself and my co-authors.